# Implementation Instruments for Developing Sustainable Tourism on Recultivated Land in the Middle Danube Flow

**Nataša Danilović Hristić \*** , **Nebojša Stefanović** and **Maja Hristov**

Institute of Architecture and Urban & Spatial Planning of Serbia, 11000 Belgrade, Serbia
\* Correspondence: natasadh@iaus.ac.rs or ndanilovichristic@gmail.com

**Abstract:** Development of sustainable tourism is viewed through the scope of planning procedure, participation of all stakeholders, and resolving possible conflicts. The methodology is based on empirical exploration and compared two case studies of the Middle Danube Flow coast segment. The common denominator, apart from the location in the same region and on the bank of an international river, is the use of recycled land for the purpose of converting it into a tourist complex. This paper has a wider theoretical background, tailored and selected for this research purpose. Ambition was expressed to answer the questions of how to carry out the strategically set tasks at the level of detailed design and implementation, what kind of interactions to expect, and if it is possible to single out key approaches and steps and form recommendations for achieving satisfactory and non-conflicting results. The authors search and look for similarities among the chosen development directions and the decisions made which can point to a common methodological framework and options for creating an attractive, profitable, and sustainable tourist product. The conclusion is that desirable sustainable tourism can be reached through careful location and content selection, choice of adequate land use, and balanced alignment between protection and development. This paper indicates the possibility of an additional step towards a joint solution, which is not only a compromise, but is valued as being of high quality and desirable.

**Keywords:** land use planning; implementation instruments; artificial land sources; recultivation; sustainable tourism; preservation; Danube

## 1. Introduction

The purpose of this research is to emphasize the approach and steps in procedure for locating tourist products in areas with natural and heritage values based on two case studies. The significance of the paper is that it contributes to the understanding of integrated planning principles for providing implementation instruments which are essential in the realization of the strategic goal regarding sustainable tourism. The background of the paper is in previously carried out theoretical research as much as practical work on the creation of planning documents with specific project tasks. Additionally, it is a part of the technical solution titled "Urban Planning as an Instrument of Implementation of the Strategic Goal of Developing Sustainable Tourism, on Case Study of the Part of The Middle Danube Flow" and adopted by the Scientific Board of the Ministry of Education, Science and Technological Development of the Republic of Serbia as important improved technical solutions and methods applied in the Republic of Serbia.

The main topic of the aforementioned is the application of an integral problem approach in the use of resources, coordinating public–private partnerships, harmonizing national and local interests, and protecting and improving the environment. The strategic goal is to promote the outstanding natural beauty (biodiversity, geodiversity, and landscape) and cultural potentials of the region (network of medieval fortresses and archaeological sites) and integrate with other attractions including nautical sport, cycling, and popular pilgrim and wine routes. A particularly important aspect is the rational use of land as the

most important and non-renewable resource reflected in the recultivation and recycling of brownfields, and artificially created locations of landfills, dumps, and embankments. In order to achieve maximum utilization and maintain the balance between space capacity and demand, the focus was on optimal dispersal of micro-locations and the protection and preservation of the environment. The result is providing conditions for the formation of a variety of tourist facilities: international passenger dock for river cruisers, two marinas, hotel facilities and complexes, sports airport, and recreational spaces and camps, with appropriate infrastructure.

The European Parliament resolution form 2021 established an EU strategy for sustainable tourism (2020/2038(INI)) [1] where tourism is recognized as a cross-cutting economic activity with a wide-ranging impact on the environment and climate, and on the EU's economy as a whole, in particular on the regions' economic growth, employment, and social and sustainable development. Since Serbia is not a part of the EU (more precisely, is in the multi-year process of accession negotiations), but the Danube is an international river and, in this section, represents the border with Romania (EU), it is important to consider and apply all recommendations. Having this in mind, all developing strategies and planning solutions are basically relying on the principles adopted by the EU: respecting general goals but taking national and regional needs and specificities into account; implementing measures and actions for more sustainable and resilient development; creating tourist opportunities that are greener and closer to nature; planning an efficient, safe, multimodal, and sustainable transport system; making a positive contribution to the economy in the areas of tourism, leisure travel, and hospitality, and preserving natural ecosystems and local natural environments, increasing the attractiveness of tourism destinations.

Choosing the Danube section as a location for research has important influence in many aspects and meanings, such as the international water corridor, the economic driver of development, the proximity of the nature reserve, the national park, and the landscape of exceptional features, including the cultural–historical route with important cultural monuments. On such a location, there is some pressure from municipal authorities and private investors to take as many as possible advantages for the construction of tourist complexes and other purposes in the domain of tourism. On the other hand, there must be a limited and purposeful approach in engaging potential sites, taking care about meeting micro-locational conditions and needs, and whether contents and capacities should be distributed without overlapping and multiplying. Observing this phenomenon, working on two similar development projects, the authors noticed certain key moments. These rose questions about the methodical approach in choosing the most suitable and sustainable location for positioning land uses for tourism. The role of the planner is, among other things, to satisfy individual needs, but not to endanger the general and broader situation.

Initially, during the urban planning phase, the authors observed similarities in two cases, especially in the selection of criteria for locating tourist facilities on demanding, artificially created, and recycled terrains. At the same time, the determination to create new opportunities for the development of tourism was decisive towards an integrated sustainable approach. In investigating similar studies carried out for this area as well as a wider scope of studies, the authors concluded that the majority of them are based on the study of one aspect (natural factors, tourist requirements, etc.) and mostly at the strategic level. This research relies on the theoretical base of the studies conducted earlier in terms of scope and topic, but in itself it represents a unique work, which cannot be compared with the previous ones, but will serve for future research and comparison of the achieved results. In this sense, this paper has the ambition to provide novelty by analyzing the methodology of detailed planning with all organizational requirements and technical obstacles but dedicated to the goal of sustainable tourism creation. Moreover, the objective is to investigate the implementation of the planned program items and cover all topics that are imposed within the planning process. The authors combine multidisciplinary knowledge from different fields such as the creation of a tourist offering, protection of ecological and cultural assets, and urban and architectural design, but always with the

premise that sustainability is the only satisfactory solution. There are several research questions to answer including: What kind of connection and interaction between spatial and urban planning and tourism development exist? What is the role of the planner? What are the methodological advantages and disadvantages during planning and setting implementation instruments? Is it possible from noted similarities and differences to highlight the key achievements and point to a good practice experience?

The structure of the paper begins with the theoretical background. The literature review refers to the relevant research thematically close to the topic and will be compared with findings of the case studies later in the discussion section. The methodology context precedes the description of the scientical approach and wider area of the case studies. In Results, the authors minutely analyze procedures of the urban planning document development for two chosen locations and situations. In the conclusions, the authors underline the sensitivity of the interpolation of the touristic facilities in the natural and historical surroundings. They assert that participation and cooperation between professionals of different specializations during the planning and designing process is essential and very welcome, as well as the mutual understanding of different stakeholders involved. The findings can be used as input for decision making and methodology upgrading followed by future research of other case study models. The conclusions discuss lessons learned and the impact of the process. Despite urban planning being quite common as an instrument for exploration and arrangement of the location, the contribution of this paper is to focus on the description of phenomena within the contexts, aiming to use experiences and compare them with other similar situations, demands, and locations.

## 2. The Theoretical Background

### 2.1. Literature Review for Sustainable Tourism

The term of sustainable tourism refers to the principles of sustainable regional development, founded on balanced territorial, social, and economic development, competitiveness and innovations, improvements in accessibility and communication, protection of natural resources and cultural heritage, and mitigation of the negative environmental impacts, achieved by all available human and technical means [2]. Danilović Hristić et al. gives remarks about the benefits of an integrated planning process in order to "make tourist product portfolio richer, with better distribution of tourist in the destinations . . . necessary to determine all different and overlapping of sectoral views and policies, starting from tourism, ecology and sustainability, protection zone's restrictions, resolve potential conflicts and define positive aspects." [2] (p. 14). León-Gómez et al. analyzes a great number of articles published to date on the effect that sustainable tourism development has on the overall long-term progress of the economy and provides a framework for further research in this field [3].

The overlapping of different types of tourism, points of interest, and suppliers with public–private partnership (PPP) as a model, can successfully lead to destination development and adequately answer the increasing market demand [4,5]. Well-organized tourism management and hospitality, balanced with local potentials in addition to generating popularity of some touristic regions could give rise to its satisfying occupancy in all seasons [6]. Opposite to this condition is the threat of "overtourism" occurrence as the negative effect, when the quantity of visitors degrade the quality of the experience [7,8]. There are a number of scientific papers about the river cruising industry [9–13] and regarding the Danube River as an important passengers' corridor with rich and intense natural beaty and a historical heritage along its course [2,14–20]. Authors Erfurt-Cooper, Tomej, and Lund-Durlacher and Jones concentrated on cruising as a tourist option; Dragin, et al. and Dwayer and Foreyth gave an overview of possible economic aspect and others, among them Renko, and Pestek, Mirea and Nistoreanu, Linnerooth-Bayer and Murcott, Sommerwerk et al., Štetić, and Pókó elaborated on the protential of the surroundings of the Danube River basin and connections with cruise routes and lines. In order to support this, Pókó observes that "River cruise tourist can be involved much more in specific programs such as: leisure

tours, itinerant discovery tourism, scientific trips for ornithologists, archaeologists and researchers, ecotourism, balneary tourism, nautical sports, fishing etc. Metropolitan guests are looking for an insight into the rural landscape, the life of the local people . . . the attractiveness of its banks as diverse landscapes, and the rich cultural-historical heritage of the nearby towns should be pointed out." [20] (pp. 38, 45). Some other papers concentrated on the wide range of offerings, eco- and rural tourism, and cycling to wine routes [21–23].

Other important topics related to the sustainability which were researched and well represented in literature are those edited by Richards or Smith, about the cultural tourism in general, and connectivity to cultural landmarks such as historical towns, castles, and monasteries with the possibility to present and preserve heritage simultaneously [24–26]. Destination choice depends not only on leisure, recreation and fun, but also on educative and cognitive opportunity. At the same time, identity of the place is excellent for branding, it gives a great impulse for development, and strongly influences the frequency of visiting and destination loyalty [27,28]. Simpson contributes with a discussion about the role and participation of community, stakeholders, and frameworks within their activity which can occur in sustainable tourism development [29]. More details about competition in the marketplace, entrepreneurship, both small and large businesses, the development of new products, creative economy, and service innovations elaborated in papers by Ratten et al. and Tajeddini will be useful in order to observe roles, tasks, scales, and levels of collaboration as well as common benefits for all participants [30–32]. For more detailed observations about the Danube basin and Carpathian Region potential, refer to Popović et al. [33].

### 2.2. Literature Review for Planning and Designing Methodology

Stefanović et al. discussed in detail a methodological framework and models of implementation for the integrated and sustainable planning in special purpose areas [34–37]. This research provides a connection between strategic planning for tourism purposes and procedures for creating and evaluating spatial and urban plans as necessary implementation documents.

For the topic of the Danube from the aspects of nature: biodiversity, environmental protection, climate change or planning in national parks, consult research by Bănăduc et al., Kovalenko et al., Strat et al., Vulevic et al., Dragićević et al., and Pihler et al. [38–43]. Regarding economical and geo-political significance and cross-border cooperation, Ágh and Kézai gave contributions in their papers [44,45]. The planning of spatial development which form the scope of crucial European strategic documents is observed by Maksin [46], and some guiding methods for improvement of the local urban planning process, including transparency and participation, is studied by Graovac et al. [47]. Since this paper goes beyond the planning phase and will touch conceptual design as an important topic, the authors paid attention to the statements of the previous research such as the importance of architectural design for positioning hotels and other touristic facilities. Mustapić and Vlahov [48] argued: "Design in architecture has always played a significant role in tourism, especially in hotel industry, where it has become one of the key factors in positioning hotels and influencing business performance . . . Innovativeness, unique architectural aspect, specific design, specificity, and personalized service are just some of the prerequisites a hotel should have to be able to meet the desires of present-day tourists. Increased people's ecological awareness brings a special contribution to this aspect as it directly influences the hotel project and hotel management by applying more and more diverse eco trends. When building hotels, certain eco principles as well as more energy-efficient materials which, apart from degrading the environment by quite minimally, also lower the overall costs of energy are now often taken into account. The concern for the environment represents a highly powerful marketing tool and 'green' is becoming a synonym for success in the hotel industry" (pp. 1, 22). Kátay and Kiss [49] have an opinion about the added value that the river bank of the Danube gives to a hotel's structure, and the importance of the orientation of rooms in order to have view. Danilescu [50] contributed the paper about floating structures

for housing and leisure along the Danube, which give people opportunities to live, work, and play on the surface of the water.

## 3. Methods and Materials

### 3.1. Methodological Approach

The main method of this research is case study [51,52], observing the phenomena within context of the study area, limited on the Middle Danube Flow region in Serbia and two micro-locations within. The authors chose a case study methodology as the most applicable and relevant because it gives the possibility to observe the process on the examples. In addition, two comparative studies within a close time range and similar conditions regarding the relationship between the purpose of the new planned structures and natural and historical location context give the opportunity to compare results.

From the researcher's perspective, methodological suitability of the case study derives from the nature of the phenomena to be empirically explored and investigated [53]. A strength of the method is in the fact that it allows tailoring procedures to the research questions. The weakness of this approach is the necessary effort of adopting the position of an objective researcher who collects facts and interprets them, since researchers were already involved in the process of preparation and development of both urban plans. This implies the wide range of decisions concerned with multiple sources of evidence, data analysis, validity, and reliability. Multiple case studies should follow a replication, not sampling logic, meaning two cases are included within one study because researchers predict that similar results will be found [54]. The methodical workflow shown in Figure 1 gives insight into the research process from starting idea to finalization with conclusions.

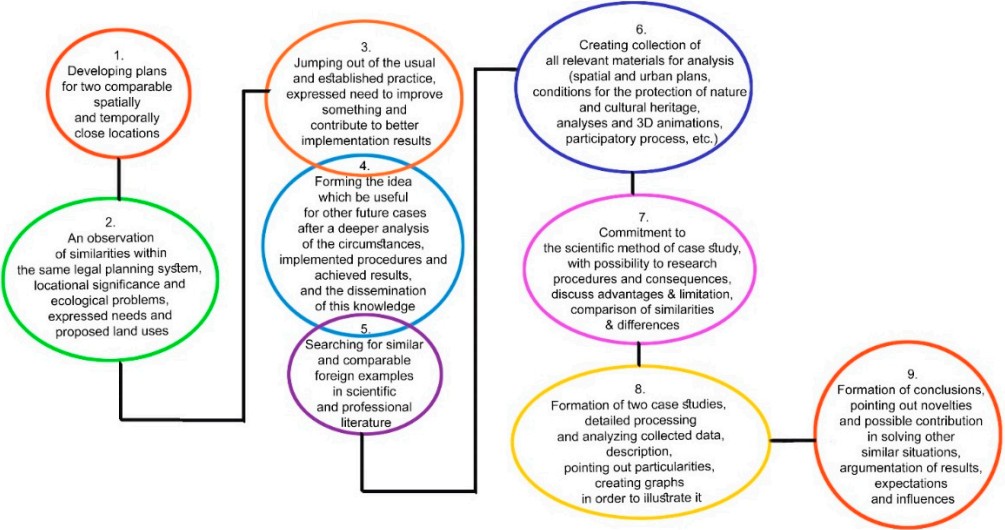

**Figure 1.** Illustration of the methodical workflow (source: Authors).

The paper consists of an introduction, theoretical background with literature review, and an explanation of used methods and materials with a description of the wider study area. In the literature review, according to the main topic and case study location, the authors provide a selective but extensive relevant review of previously conducted international research, both theoretical or based on similar case studies, related to the topics such as river potentials and sustainable tourism. The special emphasis is on choosing studies that consider issues of the fitting into the surroundings and analyzes impact on the environment. The results provide analysis for two cases/locations and comparisons. For each case study, there is an overview of the characteristics of the location, from the aspects of protection, followed by a synthesis of various conditions, limitations, benefits, and capacity for designing representative and specific tourist facilities. Since the development of urban plans preceded, thanks to the established planning practice, a large number of

opinions and conditions of competent institutions were collected and numerous analyses of the situation on the ground were carried out. As well as participation of interested stakeholders, expressed needs and real possibilities for realization of tourist capacities were considered. The discussion considers the process from the initial idea to the final result, pointing out the most important moments and stages, and comparing them with references and examples. The paper ends with key conclusions and recommendations for further research.

*3.2. The Study Area*

The Danube is about 2850 km long, and officially is the second longest river in Europe originating in Germany with inflows through 10 countries before pouring into the Black Sea. The European Union recognized this corridor as very important, one of nine multimodal networks of trans-European traffic corridors (TENT) [2]. As a border between countries, the Danube is an excellent resource for establishing regional cooperation and project proposals, such as "Attractive Danube" which improved capacities for enhancing territorial attractiveness of the Danube Region (in the period 2017–2019), as a part of the wider "Interreg Europe Programme" which shares innovative and sustainable solutions to regional development challenges [55].

Through Serbia, the Danube runs about 588 km, and is divided into three sections, Upper, Middle, and Low Flow. The Middle part, from Sremski Karlovci to Golubac, with a total length of about 190 km, includes urban centers, a picturesque landscape, areas of significant natural characteristics [56], archaeological sites from prehistoric times and the Roman period as a center of the Empire's province, Moesia Superior, and medieval fortresses on centuries-old borderlines between civilizations of Austro-Hungarians and Ottomans. In addition to the nautical route with plenty of marinas and piers for boats and cruisers, a network of roads and highway corridors, and the international E4 European long-distance path and cycling Euro Velo 6 route follows the river valley and connects major attractions. Wine tourism with gastronomy is the fastest-growing branch, being very trendy and posh, and receptive for sharing experiences on social networks. All these routes create a dense network incorporated in a spatial layout of various points of interest inside of the destination and a rich variety of manifestations, creating numerous unique itineraries as shown in Figure 2. In recapitulation, the most popular aspects of tourism in the region of the Middle Danube Flow are:

- cultural tourism (visiting urban centers, archaeological sites and fortresses, museums, rich treasure troves, cultural manifestations);
- nautical tourism (leisure time by or on water, sailing, and especially international cruising tours);
- wine and gastronomy tours (visiting vineyards and wineries, and wine tasting);
- pilgrim tourism (visiting monasteries, religious places, and celebrations);
- eco-tourism (bio- and geodiversity, enjoying natural beauty: forelands, river islands, gorges, surrounding mountains, swamps, sandy terrains; bird-watching, walking, cycling);
- hunting (fishing, visiting hunting grounds);
- congress tourism (venues for symposiums, team building programs);
- tourism connected with sports, recreation, and leisure (on river banks and beaches, different sport events, and competitions).

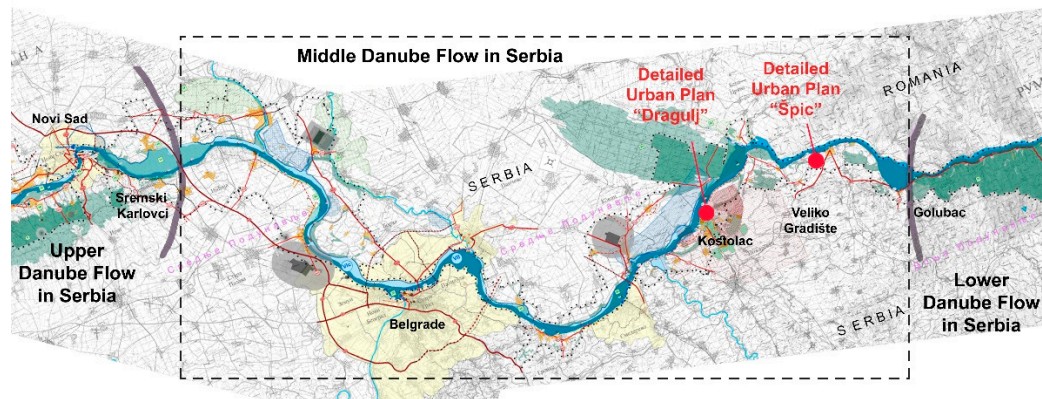

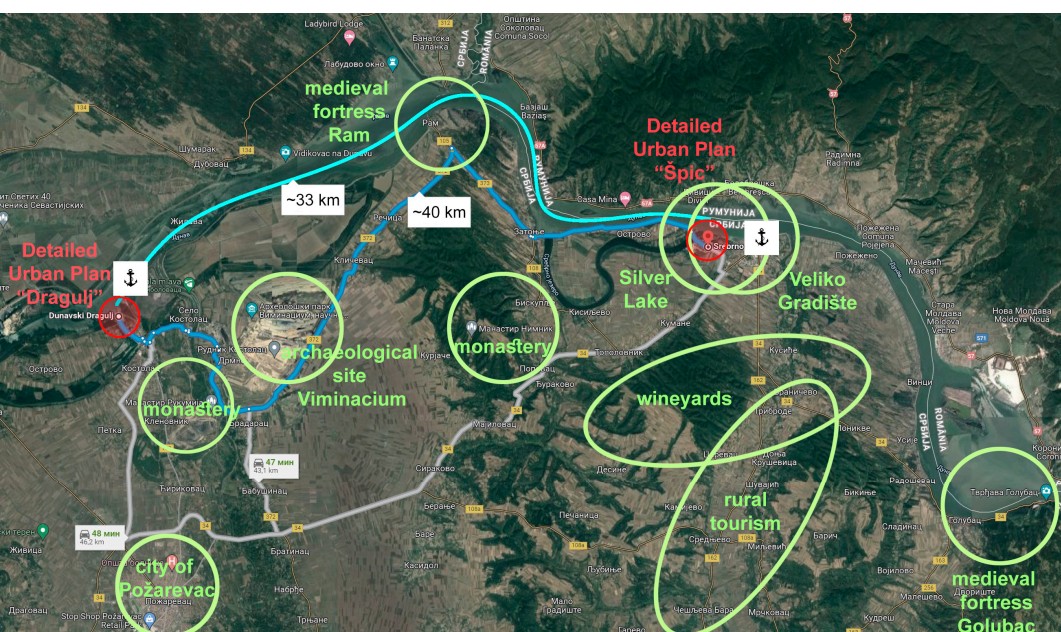

**Figure 2.** The study area of the Middle Danube Flow with case study micro-locations (source: the authors, created on the basis of the spatial plan for the area of special purpose of the international waterway E 80-Danube, Pan-European corridor VII, and Google Maps).

To fulfill the quality of visiting and staying in the region, there is a constant need for accommodation facilities of all categories from camping sites to luxury hotels. Having in mind trends and the expected increase of domestic and foreign visitors, there is an urgent need to reassess existing capacities and plan new locations, which takes into account all limitations resulting from the protection of natural and cultural and historical values, rational dispersion and concentration of tourist offerings, and postulates sustainable development. However, an integrative planning approach in the detailed positioning of facilities including passenger piers, marinas, and accommodation is much more complicated In precepting at part of the Danube coast as a whole, administrative borders between municipalities, and also international cross-border influences must still be taken into account.

In the scope of the Middle Danube Flow, the authors have chosen two micro-locations, in Kostolac and Veliko Gradište municipalities, located from each other about 40 km inland and about 30 km by waterway (stationary km 1094 + 723 to km 1068 + 200). Both locations fit the goals of the Tourism Development Strategy of the Republic of Serbia 2016–2025 [57] and are recognized in the Strategy for the Development of the Republic of Serbia's Water Transport from 2015 to 2025 [58] as a perspective for development. In addition, on the level of strategical planning [59,60], the Danube as a resource with surroundings is considered as valuable with remarks about future development and capacities. Moreover, both micro-

locations belong to the category of artificially created land, as a consequence of the filling of soil, ash, and silt, and the formation of embankments along the river as shown in Figure 3.

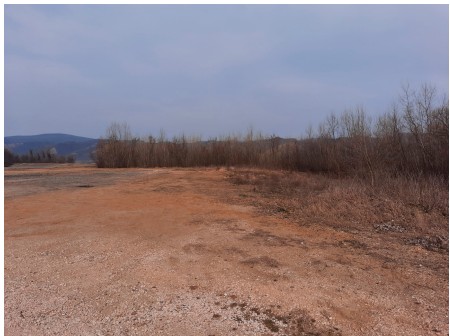
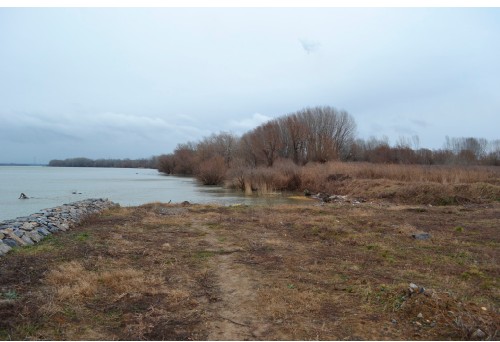
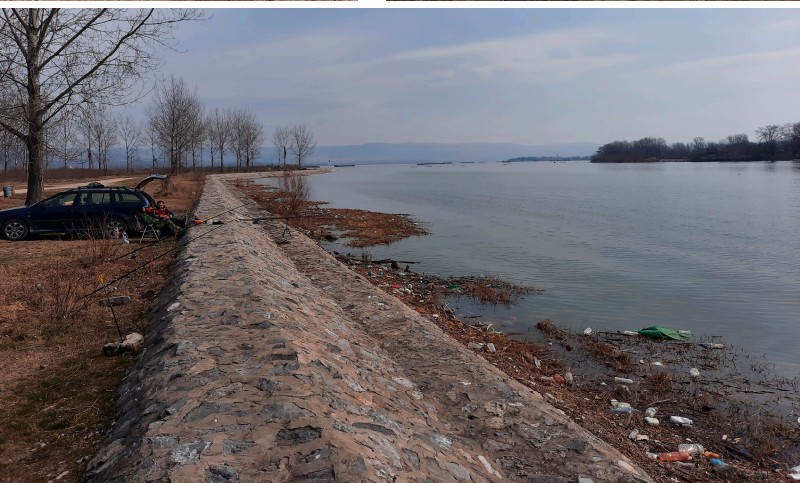

**Figure 3.** Existing condition on sites: brownfields, artificially created locations of landfills, dumps, and embankments by the Danube River in Kostolac and Veliko Gradište (source: the authors, 2018–2021).

The legal framework for the plan's hierarchy has two levels: spatial or strategic planning covering state, regional or municipality areas, and urban planning devoted to the local level and directed towards the implementation. The strategic decisions should be paired and agreed upon at all levels and finally elaborated on the local level. After making a decision on the higher spatial level, the next step is to create more detailed urban plans [61,62] that consider ideas, demands, and possibilities, fulfill all conditions, respect limitations and resolve conflict situations, and finally, allow implementation of tourist capacities. Regarding the conditions for realization, limitations, consents, and permits, these should be obtained on all levels, depending on competencies. The role of the urban planning expert is to perceive all circumstances on a specific area, and examine and propose solutions including requirements and possibilities. The position between different stakeholders (local authorities and decision makers, institutions that issue conditions and consents, professional bodies, private investors, and citizens) and transparency of the procedure requires mediation and negotiation skills.

## 4. Results

### 4.1. Analyzing the Location ''Dragulj'' in Kostolac

The detailed regulation plan for location "Dragulj" ("A Jewel", as a name of the restaurant on the most prominent point), was initiated and financed by the city of Požarevac in 2018 and adopted in 2019 [61]. The long-term goal was to promote nautical tourism in the first place and apply sustainable development as a principle. The main short-term goal was to design the passenger port for river cruises, in order to provide debarkment

for tourists, and organize visits to archaeological sites of the Roman provincial capital and military camp "Viminacium" only 5 km away. The plan covered about 53 ha of land surrounded by water form three sides, between the main stream of the river Danube, its lateral branch and canal (as a function of a nearby thermal power plant). This land is partially recultivated brownfields and a previous ash dump, so it was very important to perceive all environmental issues. At the same time, sustainability is partially fulfilled by the reusing and recycling of the land, instead of occupying the natural coast and disturbing its biodiversity.

The passenger port takes an area of about 3845 m$^2$, with a building complex of 500 m$^2$ containing accompanying offices of the operator, passport control, tourist information point, restaurant, rent-a-car, souvenir shop, medical aid, and toilets. The port provides for the disposal of garbage from the boats, the possibility of tank water, and obtaining food product supplies. In addition, in front of it is situated a car park for buses and induvial cars. It started to function in the spring of 2022 as the 7th functional pier on the Danube in Serbia, therefore, we are still without data regarding the number of dockings and passengers during the first season.

In addition to the passenger port, various other land uses connected with the development of tourism were incorporated into the plan. Some of them are very specific with particular regulations and requirements to fulfill and coordinate with remaining uses. Among them are:

- a marina, designed for 100 boats, according to standards for 3 anchors (equivalent to hotel category classification), with all necessary facilities and services;
- a sport airport for small airplanes, followed by an administrative building with flight control equipment, hangar, workshop, air club, medical aid, and fire service.

The plan provided capacities for improvement of the accommodation on the site. It was calculated by considering very popular short breaks, up to 2 or 3 days, during the weekends or holidays. To accomplish this goal, the existing guesthouse and restaurant can be reconstructed and upgraded to the upper category; existing housing within the plan (mostly private weekend houses by the river's branch) can accommodate B&B guests using Booking.com, Airbnb or similar popular online reservation systems. The craft production factory has the possibility of changing purpose and transforming to a hotel complex of 7100 m$^2$ gross building area (GBA).

About 50% of the plan area (approximately 22 ha) is intended for landscaped greenery, public land for leisure, and diverse sport fields, such as terrains for football, basketball, tennis, beach volleyball, badminton, skate park, jogging and cycling lanes, children's playgrounds, picnic areas, etc. For tourists devoted to nature and eco-tourism, the plan provides a camping site area with all the necessities. The cycling path that surrounds the area is an addition to the main Euro Velo 6 route. On highlighted points by the river Danube, planners designed observation desks and viewpoints as attractions.

The main planning challenges in order to develop sustainable tourism were to resolve conflict situations and different demands, especially between the passenger port and marina on the river bank and the direction of the airport runway approach between them. The protection of the nature and quality of soil composition were important topics, bearing in mind that the area which was artificially filled and recultivated by time became a part of the ecosystem. All planned capacities for tourism demanded satisfying traffic (streets and pedestrian promenade, parking spaces) and infrastructure supply (running water, sewage, electricity, heating, gas installations, concerning energy efficiency), and embankment for flood protection. The most important task was to perceive and take advantage of the space, making a combination of various tourist products for different users' preferences, but to keep balance between quantity and quality as shown in Figure 4. For this reason, planners tried several scenarios to test mutual positions, relationships, and interactions of proposed land uses in the space.

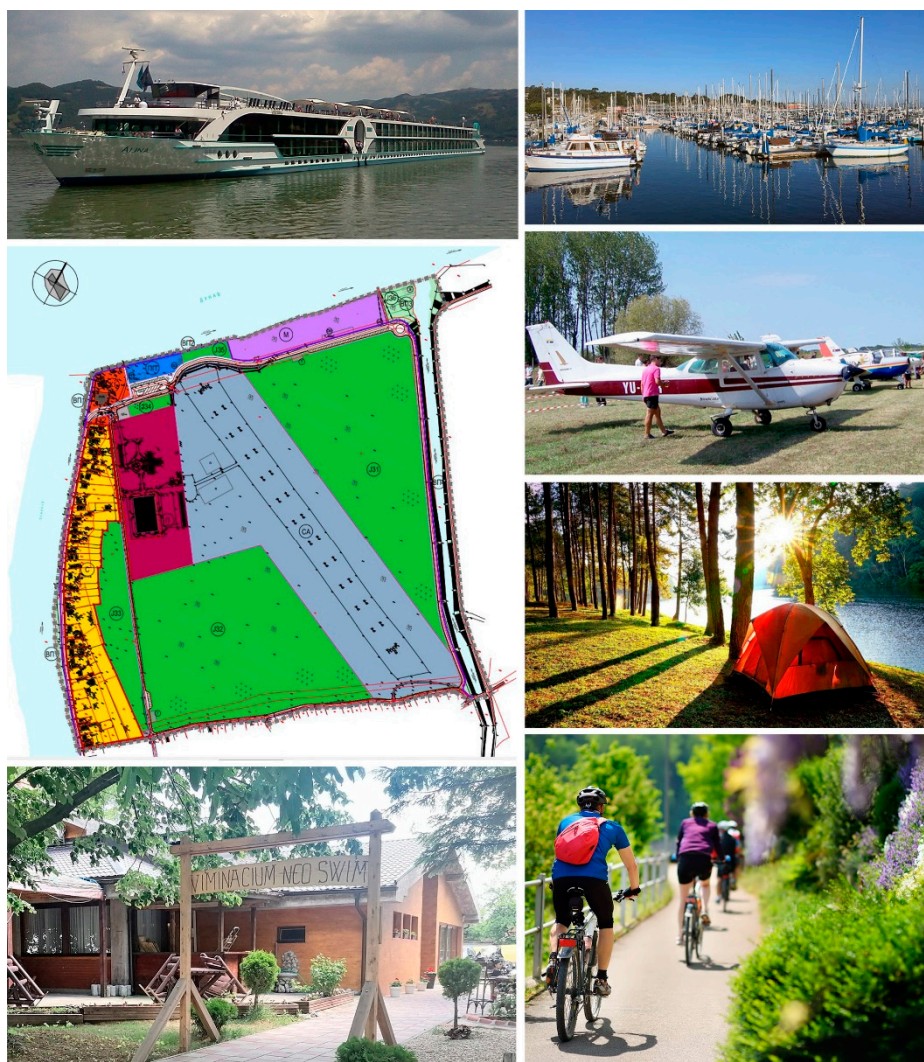

**Figure 4.** Detailed urban plan for "Dragulj" in Kostolac (source: authors; plan developed by the Institute of Architecture and Urban & Spatial Planning of Serbia—IAUS, other pictures publicly available on the Internet).

### 4.2. Analyzing the Location "Špic" in Veliko Gradište

The detailed regulation plan for the location of "Špic" ("Peak" which is an association for the shape of the land area), located on Danube, directly next to Silver Lake, was initiated and financed by a private investor and adopted by the municipality of Veliko Gradište in 2021 [62]. Their mutual interest was to use artificially created land by the bulwark that separates lake from the river flow (lake was created by enclosure of the Danube's lateral branch).

Stones which have been thrown and spills of sludge and other materials as a result of the cleanup of the Silver Lake forms a breakwater with a lighthouse and winter shelter for small boats. The plan covered an area of about 18 ha. The status of the land is public, but there is consensus that is not useful in the present state and may be privatized through the process of auction by collecting bids. The location is technically very demanding and complicated for realization, but at the same time, unique and unrepeatable, bearing in mind that it is located right on the Danube coast, surrounded by water and with beautiful views of the river and the opposite side. On this point, the Danube represents a natural border between Serbia and Romania. Silver Lake itself is a popular touristic resort, with an organized beach, surrounded by apartments for renting and small hotels, a fun aqua-park, sport fields, camping sites, and restaurants. There is a hotel of high categorization, however, similar accommodation is in deficit, therefore, this was a reason for the initiative to reuse

this 'empty' location and situate a luxurious minimum 4-star hotel with a condominium complex in the annex.

The structure of the hotel accommodation units and the number of beds depends on the category of hotel and the prescribed standards, and for the purposes of the planning, it is calculated that about 40% of capacity will consist of double rooms, about 50% of the rooms will have a king-size bed (average area 30 m$^2$), about 3% of the rooms will be adapted for persons with disabilities, and 7% will occupy apartments and family rooms (average area around 65 m$^2$). The estimated total number of units is approximately 200 and the achieved gross building area (GBA) of the hotel is about 20,000 m$^2$. In addition, the hotel should include an entrance hall, restaurants with kitchens, utility spaces such as a laundry, a spa with pool, recreational facilities, as well as a space that will be suitable for the development of congress tourism and a venue for weddings and other events. Thanks to the shape of the location and its position between the two water surfaces, the structure is designed with two-sided orientation, meaning that all rooms will have balconies with a direct view to the river. The maximum height of the hotel building is ground level and four upper floors. The condominium annex is oriented towards the Danube, with protective greenery towards the access road and parking. The structure of the apartments can be made up of units of different sizes, from 60 to 120 m$^2$ of surface. Part of the ground floor of the building can be used for other commercial purposes within the complex. The planned gross building area (GBA) is about 8000 m$^2$ organized into eight units, with a maximum height of ground level and two floors. The plan proposed to blend and adjust the appearance of the architecture into the ambience, but also to exploit the location's highlights and accentuate distinguishing features. It is prescribed to use a contemporary architectural expression and materials, but to avoid large reflective surfaces on facades in order to protect bird habitat and their migratory routes.

In addition to the hotel and condominium structures, the plan gives the opportunity to supplement the tourist product with a marina for smaller yachts and sailboats, shared spaces in surroundings with greenery, sport terrains, walking and cycling paths, observation platforms, pontoons/floating platforms on the river for water sports, a pool area, restaurant gardens, sun terraces, fishing points, and additional accommodation units. The boat mooring capacity of about 50 vessels meets the requirements; its location is off the waterway and it is protected from the effects of waves, winds, river drifts, etc. Special attention is given to access and the parking for the whole touristic complex.

The main planning challenges, in order to develop sustainable tourism, were to provide infrastructure supply and sewage drains, embankment for flood protection, overcome the demanding soil composition and high level of underground water, and implement all measures for the protection of the internationally important wild bird habitat (IBA—Important Bird Area). The most important task was to take advantage of the unused but attractive location and resolve potential conflicts between demand and real space capacity. For that reason, the plan went beyond mandatory scope and produced a conceptual 3D design of planned structures to demonstrate and attest results. In order to preserve public access to the most prominent point of the location, its peak with the viewpoint, planners decided to locate the hotel on the more technically complicated elongated part, and condominiums on the wider section. By this, the luxury complex still has a dose of openness and invitation for visitors that are not its guests, but are interested in enjoying the unique design and landscape (Figure 5).

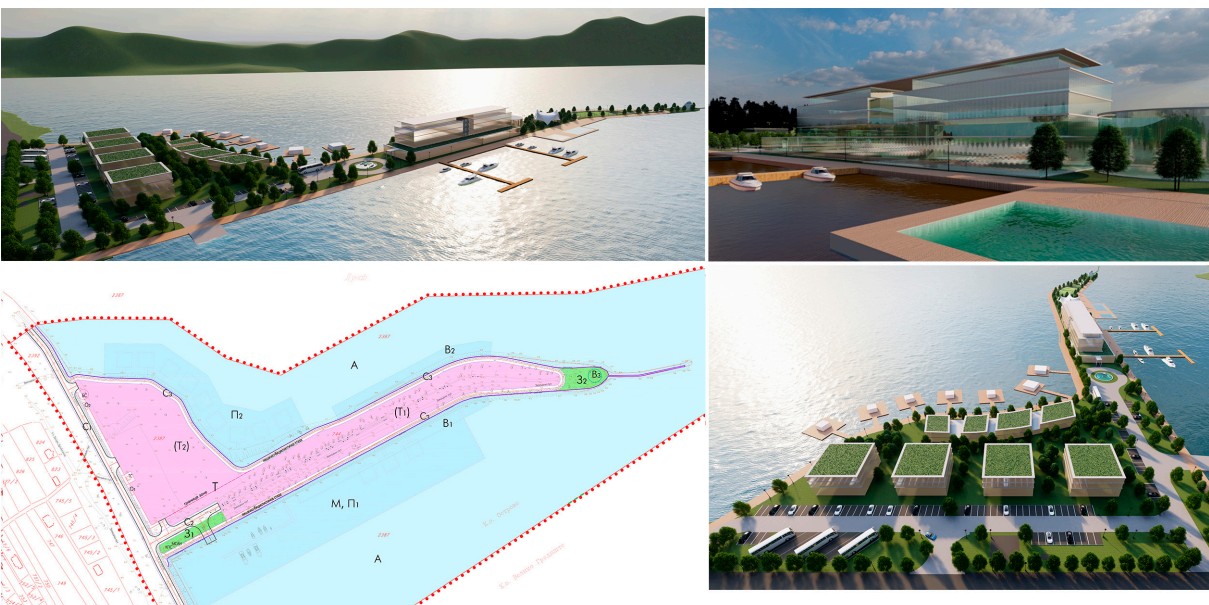

**Figure 5.** Detailed urban plan for the touristic complex "Špic" in the municipality of Veliko Gradište (source: authors; plan developed by the Institute of Architecture and Urban & Spatial Planning of Serbia—IAUS; conceptual design by architect G. Babić, Urbanpro inženjering doo).

*4.3. Comparison of Two Locations*

Two potentially prospective touristic locations, analyzed as case studies, have some similarities in common, but also differences as shown in Table 1. These should be compared with the goal to conclude how spatial and urban planning influence creation of the tourist product and what are the implementation instruments for sustainable tourism. Expected effects of developing and adopting detailed urban plans are in the scope of the realization of all planned facilities and in promoting touristic potentials of the region, completing the touristic offerings and increasing visits. The methodology of the planning highly required an integrated and comprehensive approach in resolving conflicts and fulfilling the project tasks.

First, it is important to emphasize the hypothesis that regardless of the closeness of the locations and the overlapping of some land uses (hotel and marina facilities), there are no obstacles for functioning separately or together in the region of the Middle Danube Flow. The positions of the processed areas gravitate to the different attractions: "Dragulj" is more determined for the archaeological site of Viminacium and cultural events in the surroundings of the city of Požarevac, unlike the "Špic" which is connected with the resort of the Silver Lake and wine and pilgrim tourism in the surroundings. The Danube, as a connecting element, provides plenty of possibilities for nautical tourism and there is still demand for establishing passenger ports, piers/docks, and marinas, in accordance with strategic level decisions. "Špic", as a smaller location and monofunctional hotel complex has no need to be primarily connected with a port for cruisers, since these tourists stay overnight on the boat, however, there is one nearby in the center of Veliko Gradište (only 3.5 km away). It is more important for this location to provide access by yachts as well as by cars and tourist buses, and that is the reason a marina is planned. More than less, the location has an ideal predisposition for nautical tourism and safe anchoring. Attractions such as the medieval fortress of Ram, Christian Orthodox monasteries, vineyards, rural (ethnic and gastronomic) touristic destinations, etc., situated between the two locations are easily reached from both. The areas are connected by local roads and the river and, above all, by the European cycling route.

**Table 1.** Comparison of planned contents and touristic products for two case study locations (source: authors).

| Plan Contents and Touristic Products | Detailed Urban Plan "Dragulj" | Detailed Urban Plan "Špic" | Similarities | Differences |
|---|---|---|---|---|
| Coverage (ha) and position, formation | 53 ha, by the river, partially recultivated ash dump | 18 ha, by the river, artificially created stone throws and spills of sludge | Danube coastline, surrounded by water and land, conversion and reuse of artificially created land by backfilling | Coverage area |
| Land use | - Passenger port for cruisers, marina, small airport, sport and leisure facilities, camping, hotels, and apartments. Share in the area of the plan: port and marina 10% <br> - airport 25% <br> - greenery and sports 42% <br> - traffic and infrastructure 7% <br> - hotels and accommodation 16% <br><br> Public and private ratio = 84:16% | - Hotel and condominiums (apartments), marina, sports and leisure. Share in the area of the plan: hotel and condominium complex 13% <br> - water surface (marina and sports) 73% <br> - greenery 1% <br> - traffic and infrastructure 13% <br><br> Public and private ratio = 87:13% | Same methodology of planning. The majority of land use is in the scope of creating sustainable tourist offerings and product. Majority of the urban land in public (state or local authority ownership) | Various, multifunctional in "Dragulj", testing of combinations through scenarios/monofunctional product in "Špic", testing of capacity and appearance of a thorough conceptual design |
| Advantages | Enough free space for various and compatible tourist facilities, proximity of archaeological site, connectivity | Prominence and visibility, proximity of touristic resort and routes, connectivity | Integrated planning method for resolving conflicts and problems in space, sustainable development as a premise, maximum use of capacity with care for quality and originality, as much as for a burden on the surroundings and ecosystem, creating new focal points and brand, protentional risks in phases of implementation regarding technical demands | - |
| Disadvantages | Ecological issues, proximity of open pit and thermal power plant | Shape of the parcel, soil composition | | - |
| Demands and technical solutions | Soil composition, flood defense, infrastructural supply, reconciliation of different technical requirements for contents (port, airport, marina, etc.) | Foundation of structures, flood and underground water defense, infrastructural supply and drain, need of alternative solutions | | - |
| Accommodation facilities (GBA/number of units) | 18,000 m$^2$ in total, about 80 hotel units and 40 B&B apartments | 28,000 m$^2$ in total, about 200 luxury hotel units and 70 condominiums | Rational use of capacities, location of accommodation | Difference in capacity and categorization. Main goal of the plan "Špic"/secondary function in the plan "Dragulj" |
| Nautical tourism | Passenger port for cruisers (500 m$^2$) and marina (3 anchors = 100 boats) | Marina for max. 50 boats | Using river as a resource for tourism development, sports on the water (paddling, kayak, sailboats) | Main goal for the plan "Dragulj"/secondary function as a supplement to the hotel for "Špic" |
| PPP—public–private partnership | City government financed plan development, majority of land is public, selected port operator is archaeological park Viminacium, accommodation facilities belong to private businesses | Private company, interested in development of hotel complex financed urban plan, the result will be change of ownership, renunciation of the state from the unnecessary | Procedure of plan development, evaluation, and verification according to the Law acts, including public hearing and participation of all interested parties | Stakeholders involved in the process are different. In DUP, "Dragulj" land stands mostly public by appointing the holder. In DUP, "Špic" land will be privatized through the process of auction by collecting bids |
| Environmental impact | Local authorities decided that there is no need for EIA as a separate document | Local authorities made the decision that EIA is mandatory, this document required extra procedures, detached from the plan, and was finally adopted simultaneously | On both cases, urban plan examined all conditions and prescribed measures for environmental protection and gained positive opinion from authorities | The procedure for EIA (Environment Impact Analyses) |

**Table 1.** *Cont.*

| Plan Contents and Touristic Products | Detailed Urban Plan "Dragulj" | Detailed Urban Plan "Špic" | Similarities | Differences |
|---|---|---|---|---|
| Economic and social impact | Revenue from port services, arrangement of space will impact on number of visitors, creating new events | Branding the wider location with a new luxurious offer | Creation of new jobs, innovation in offer, public works (especially for communications and infrastructures) | - |
| Architectural style, visual impressions | On the first line of the coast are passenger port and marina, different forms of accommodation capacities (hotel, weekend houses and apartments, camping sites) are secondary | The hotel facility stands out. The apartment complex has a more intimate character. All objects have orientation towards the river flow, with addition of the attractions on the river banks | Both plans insisted on rules and recommendations for a contemporary architectural expression, compatible with the environment, but at the same time, iconic and recognizable. Position by the river is demanding, but provides added value | - |
| Adoption of sustainable practices | "Recycling" and rational use of the land as a resource, measures for preservation of the habitat, limitation of capacities, sharing of contents in the surrounding and complementing each other, energy efficiency, green and other alternative energy use, waste management, stakeholders' cooperation, community involvement, promoting of the region | | | |

The greatest similarity is the origin of the locations, both on the Danube bank, created by human activity as artificial and recultivated brownfield land. As land is the most valuable resource, it is commendable and completely in the style of sustainability to reuse and recycle it for new purposes. Another common problem was to plan and design infrastructural capacities for all facilities and provide alternative and renewable resources of green energy as well as prescribe energy efficiency recommendations. All planned purposes and their consequences on the surroundings were perceived, evaluated, and checked by the Environment Impact Analyses (EIA). The main difference is in the coverage area that consequently affects multi- or monofunctional land use and the achieved accommodation capacity.

Public–private partnership (PPP) is another mutual issue, implemented by the same legal procedure, but with completely different results. An urban plan is a public document, with a procedure of production, evaluation, verification, and adoption that includes stages of the professional control and participation of citizens and all stakeholders. No matter who is financing the planning document, it will undergo all legal steps and needs to obtain the necessary consent of the relevant participants and institutions responsible for planning, protection, and maintenance of the space, on all levels, from the municipality to the state. It is crucial to defend public interest and goods, preserve values, and simultaneously fulfill the demand for future development. Both plans succeeded in this, using a method of interdisciplinary and integrated planning for resolving spatial inconsistencies. The expected results should be in the sphere of economic impact, empowerment, and competitiveness of the local communities, creating new jobs, branding the product and locations, and cooperation in creating an original touristic offering. Moreover, tourist organizations of the region supported ideas embedded and carried out by the planning documents. It is significant to emphasize that both plans during the phases of participation and public hearing did not have any negative remarks or comments regarding their proposals from the local authorities and state institutions, utility companies, citizens/communities or NGOs. This is an extremely rare situation where a development plan does not have any obstacles for acceptance and adoption, and this fact testifies about critical perception of the demand and possibilities for implementation.

## 5. Discussion

The topic of sustainable tourism is well represented in academic papers, describing the relationship between the tourism or travel industry and tourist destinations, and explaining circumstances under which assumptions of capacity should be adopted by destination planners [63,64]. The tourism industry, as one of the world's largest industries

and an important sector for many nations, refers to all activities related to the short-term movement of people and includes the hotel, transportation, and a number of additional industries or sectors. It is based on different travel motivators: leisure, cultural, events, business, and others. Tourist destination is connected with the spatial aspect; it refers to the geographic area or zone frequently visited by tourists. This kind of place possesses some significance, motivation or attractiveness, and natural or built value that offers leisure and amusement. The unit of space may vary in its size from country/state, region, city, resort or smaller area, which has a collection of tourism-related products. Sustainability issues should be considered on the highest strategical level when determining the development of the tourist industry as a crucial driving force and planning layouts and networks of tourist destinations.

Weaver emphasizes the utility of development standards as a means of ensuring the quality of tourism-related landscape modifications, evaluates the spatial strategies, role of zoning and (re)development, considers negative environmental and sociocultural impacts, and discusses potential tourism outcomes. Harris et al. [64] analyzes the difficulties associated with coordination and cooperation between stakeholders and policy makers involved in sustainable tourism and the limitations in adoption of sustainable practices. Zhenhua [65] even gives a brief critique of some of the weaknesses in the sustainable tourism literature, in particular, exploring issues that are often overlooked: the role of tourism demand, the nature of tourism resources, the role of tourism in promoting socio-cultural progress, and the measurement and forms of sustainable development. Finally, in order to transform research on sustainable tourism to a more scientific level, a systems perspective and an interdisciplinary approach are indispensable. An interdisciplinary and integrated approach is a crucial element in planning methodology in order to produce implementation instruments for realization of the sustainable touristic product and to remain sure that it will function without obstacles or negatively impact the surroundings. As Danilović Hristić et al. [2] concluded, "... the suitability was elaborated in form of assessment of impacts and in process of gaining spatial conditions form the list of relevant institutions responsible for environmental and heritage protection in the first place. In combination of overlapping layers of protection zones and defined restrictions with all other technical postulations, it is possible to locate the "ideal" position..." (p. 13).

As Chen and Rahman, and Richards [27,28] discussed, destination choice depends on identity and profound roots of the location, giving extra quality and additional pleasure for visitors, therefore there is a strategic determination to develop the Middle Danube Flow in Serbia based on secure foundations. An international river with all conveniences and countless possibilities has unrepeatable natural beaty (biodiversity, geodiversity, and landscape), and is a valuable ecosystem in its surroundings as well as a cultural milieu which has developed for centuries. Several studies [38–43] gave key remarks about nature values in the observed region and environmental impacts. Micro-location positioning depends on a set of criteria, conditions, and possible adjustments to requirements. According to Simpson [29], it is mostly about the framework, roles of stakeholders, and the development of new products, creative economy, and service innovations. In the discussion of Ratten and Tajeddini [30–32], there is an elaboration of the means of collaboration between stakeholders and the benefits which can be gained. The process of planning has an important component of participation, in the form of public insight and discussion, and because the topic is a public good of a unique river bank, the opinion about the project is very important, as concluded by Graovac et al. [47]. It is realistic to expect that the result of producing possibilities for new tourist products will increase competitiveness and empowerment of the community. From the planning point of view, locating and zoning have several extra aspects to take into account: distance from the other products and points of interest, connectivity and accessibility, basic technical conditions, and finally, maximization of resource use but optimization of loading with contents and visitors. Both case studies during the planning phase considered not only the basic project proposal but also combined and harmonized it with the wider public interest, and this should be

taken as a positive example. The authors completely agree with the findings of Mustapić, Kátay, and Danilescu [48–50] regarding influence of architectural appearances, the balance between the iconic, and the embeddedness with surroundings.

## 6. Conclusions

The Middle Danube Flow region takes about one third of the river length through Serbia and represents a capacitive touristic destination where natural and cultural attractions are recognized. It is to be expected that an increase of frequency of visits and dissemination of the values will initiate further development of nautical, cultural, ecological, rural, and other types of tourism. The tourism may be an excellent start for communities' progress and prosperity. Urban planning, as a discipline, provides an integrated methodology and instruments of implementation for satisfactory results and balanced growth. Regulations, sets of rules, and compulsory standards remelt into good design and have an influence on accomplishing pleasant surroundings and providing quality offerings. The sustainability, as a principle, requires a detailed and comprehensive approach to all conditions, phenomena, and requirements, with an assessment of possible impacts of the planned land uses and capacities on the area.

The initial research questions were regarding interaction between planning and tourism development, methodological characteristics of planning which affect implementation instruments, and the possibility to extract the key achievements of good practice experience as guidelines for similar situations. By comparison of the two case studies, the authors searched and looked for similarities among the chosen development directions and the decisions made which point to common frameworks and answers that can be translated into rules and recommended methodological steps.

The paper envisages connection and interaction between spatial and urban planning and tourism development on two spatially close case studies, analyzing conditions, similarities, and differences between them. The role of the planner is to resolve all possible conflicts in the space, to secure public interest, and to protect certain values. At the same time, the planner has the task of providing development and improvement in accordance with demands and needs. In this process, the planning professional relies on numerous spatial and other data, guides with procedural steps, and encourages cooperation and understanding between different levels of decision making and interest groups. These two examples represent a situation where the urban planning expert, in accordance with the local community and private sector, chooses a less convenient investment, even more, risking a piece of land to locate touristic products, but with a goal to preserve other high-quality locations on the Danube banks. Reusing and recycling of previously partially devastated, unused, and neglected space as a resource has sustainable character. Based on the experience with these two case studies, the authors ratiocinate that EIA (Environment Impact Analyses) should be mandatory for similar situations, but other documents such as Social and Economic Impact Analyses (SIA, EIA) may also be suitable and useful.

Descending from the strategic level of possibilities, ideas and proposals to the more precise and detailed level of positioning, fitting, defining spatial requirements, relationships and capacities, requires a comprehensive approach in resolving potential conflicts and minimizing negative impacts. Efforts in the perception of other components of planning, considering the sensitivity of the interpolation of new contents, combination of the land uses, mutual relations, maintaining balance with the environment, predicting the maximum of utilization and limiting it to the optimum levels, infrastructure furnishing etc., are leading to the same goal of sustainability.

The academic relevance of the paper is that it contributes to the understanding of instruments that are essential in realization of the strategic goal regarding sustainable tourism. It is a wider methodology in regards to a multidisciplinary and integral approach, combining knowledge from fields of tourism, spatial and urban planning, protection, and preservation. This integrated method equally takes benefits and disadvantages as arguments for the problem approach and resolves possible conflicts in the context of

sustainability, with assessment on how to harmonize specific spatial demands. In order to connect spatial and urban planning with sustainable tourism, it is important to cover topics starting from procedures of developing, evaluating, and decision making of plans, simultaneously with analyzing regional characteristics and potentials, applicable types of tourism as a desirable product, and finally, to point out key stakeholders and locations. The authors are following a matrix of chronology from a strategic to detailed phase parallel with the levels of competence in the public sector (from state to municipality level) and private sector, and give main intersections regarding the focal points of interest. The research involved several activities, consisting of reviewing existing data included in strategical documents (strategies and plans), environmental impact assessments (EIA, respecting of restrictions and conditions), demands for creating and promoting touristic products, and finally, technical solutions on an urban planning level. It uses results of previous analyses, such as those about the expansion of the river cruising industry and the spatial planning instruments, with the goal to indicate correlation between demands and steps of the implementation of the strategic goal.

In order to emphasize the importance of the balance between the growth and development on one side and protection of natural resources and cultural values on the other, relying on previous analyses, this paper gives a path for the successful resolving of hypothetical conflict. It provides ideas for content selection, rational distribution, and land use, with a maximization of diversity and quality of the touristic offerings.

The scientific contribution of this paper is in the sphere of empirically carried out analyses regarding correlations between diversity of options for creating attractive and profitable tourist products and demands for sustainability. Using the case study from practice, inspired by achieving alignment between the protection and development, this paper argues about planning techniques and methods as primary implementation instruments for location selection, choice of adequate land use, organization of contents, and technical requirements. This paper indicates the possibility of an additional step towards a joint solution, which is not only a compromise, but is valued as being of high quality and desirable.

The limitation of this research is lack of similar case studies to compare with. The current challenge is the fact that the development and operation of new planned facilities are still in progress, so there is still no clear evidence regarding what will be the real effect, which arises the reason to follow and monitor the progress and impacts in the future. The lessons learned may be useful for other professionals and the description of the experience may lead to further research: methodology upgrading, dispersal and diversity of tourist destinations on the same route or inside the same region, studies on environmental, economic and social impacts, resolving spatial overlapping of protection zones and tourism interest, creating products and adjustments to different visitors, and maybe the most important, raising questions regarding the sustainability and critical level of exploitation of locations. Based on this research, it would be very useful to continue with investigations on similar locations and occasions within the region of Middle Flow, but also the wider area, especially the Upper and Lower Danube Flow in Serbia, and to take into account cross-border influences and collaborations with Hungary and Romania. This inclusive approach will benefit with results about the output of overall sustainable tourism development and effects of the implementation of strategic goals.

**Author Contributions:** Conceptualization: N.D.H.; methodology: N.D.H. and N.S.; formal analysis: M.H.; investigation and sources. N.D.H.; data curation: M.H.; writing—original draft preparation: N.D.H. and N.S.; writing—review and editing: M.H.; visualization: N.D.H.; supervision: N.S. All authors have read and agreed to the published version of the manuscript.

**Funding:** Funds for the realization of the research shown in this work are provided by the Ministry of Education, Science and Technological Development of the Republic of Serbia, grant numbers 451-03-68/2022-14/200006 and 451-03-68/2023-14/200006.

**Institutional Review Board Statement:** Not applicable.

**Informed Consent Statement:** Informed consent was obtained from all subjects involved in the study.

**Data Availability Statement:** The data presented in this study (in Serbian language) are available publicly or upon request from the corresponding author.

**Acknowledgments:** This research is a part of the technical solution titled "Urban Planning as an Instrument of Implementation of the Strategic Goal of Developing Sustainable Tourism, on Case Study of the Part of The Middle Danube Flow", in the category M84—Important improved technical solutions, methods applied in the Republic of Serbia used in at least one institution, adopted by the Scientific Board of the Ministry of Education, Science and Technological Development of the Republic of Serbia on 31 May 2022.

**Conflicts of Interest:** The authors declare no conflict of interest.

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
