# Peer review of "Implementation Instruments for Developing Sustainable Tourism on Recultivated Land in the Middle Danube Flow"

_sustainability, doi:10.3390/su15097724_

Round 1
Reviewer 1 Report
I find the theme of the article particularly interesting, as it deals with the balance between growth and sustainability in tourism. Both concepts can and should go hand in hand.
In general terms, the paper needs to be provided with a much more engaging introduction and conclusions. Also, even the background needs to be expanded. On the other hand, the case study seems interesting and well described (although sometimes too descriptive), but the overall article loses strength because of the introduction and conclusions. I think that if the authors make an effort in the first and last sections above all, the work will gain more relevance.
I provide some suggestions that could improve it based on major revision of the manuscript:
Introduction
The introduction is too brief. It contains the objective and some summary conclusions. However, a proper introduction should be much more attractive. In my opinion, it should be a guide to what we are going to find in the article and make readers want to approach the topic. A research gap and clear research questions should be properly stated. The authors also explain the structure of the article, which I think should be at the end of the introduction.
Methods and Materials
At the beginning of this section the authors state "The main method of the research is case study" (line 46). These are statements that must be referenced to be substantiated.
The academic and scientific contributions I would put in conclusions. In addition, as a summary in the introduction.
The Theoretical Framework
There is a need to expand and create sub-sections that put the research in better context.
On line 140 the authors indicate "papers by numerous authors" which authors? Give some examples.
The Study Area and Analyzing the locations and options for touristic products within them
This sections seems to me to be very well explained. The images made by the authors are valued. The comparison between the two areas is very clear. I would give the sections a more attractive title as it is too descriptive.
Discusion and conclusions
The discussion seems to me to be adequate with the literature review conducted and the comparisons proposed.
The conclusion should be rewritten to reinforce it and make the contributions of the paper more visible. Answer research questions posed in the introduction.
I thank the authors for the opportunity to review their work.
Reduction of some long sentences to become more concrete.
Author Response
Answers to the reviewers
Dear Reviewer 1,
First of all, thank You for your kind words, effort to review our paper and extremely useful comments and remarks. You gave us very clear and detailed inputs how to improve this article and we really appreciate your help. Sincerely, we hope that we understood everything correctly and accomplished expectations. We tried to combine all remarks form 3 reviews, and all corrections and additions are implemented in new version of the paper, that we are uploading as an attachment.
I find the theme of the article particularly interesting, as it deals with the balance between growth and sustainability in tourism. Both concepts can and should go hand in hand. In general terms, the paper needs to be provided with a much more engaging introduction and conclusions. Also, even the background needs to be expanded. On the other hand, the case study seems interesting and well described (although sometimes too descriptive), but the overall article loses strength because of the introduction and conclusions. I think that if the authors make an effort in the first and last sections above all, the work will gain more relevance.
I provide some suggestions that could improve it based on major revision of the manuscript:
Point 1: Introduction
The introduction is too brief. It contains the objective and some summary conclusions. However, a proper introduction should be much more attractive. In my opinion, it should be a guide to what we are going to find in the article and make readers want to approach the topic. A research gap and clear research questions should be properly stated. The authors also explain the structure of the article, which I think should be at the end of the introduction.
Response 1: Accepted and corrected. We changed and expanded Introduction section.
Point 2: Methods and Materials
At the beginning of this section the authors state "The main method of the research is case study" (line 46). These are statements that must be referenced to be substantiated.
The academic and scientific contributions I would put in conclusions. In addition, as a summary in the introduction.
Response 2: Accepted and corrected. We added references about case study method, with some extra explanations and additions about methodological process. Contributions and summary moved to suggested chapters.
Point 3: The Theoretical Framework
There is a need to expand and create sub-sections that put the research in better context.
Response 3: Accepted and corrected. Paper has new order (according to comments of Reviewer 2) with some new subtitles.
Point 4: On line 140 the authors indicate "papers by numerous authors" which authors? Give some examples.
Response 4: Accepted and corrected.
Point 5: The Study Area and Analyzing the locations and options for touristic products within them
This section seems to me to be very well explained. The images made by the authors are valued. The comparison between the two areas is very clear. I would give the sections a more attractive title as it is too descriptive.
Response 5: Accepted and corrected. As Reviewer 3 suggested structure of titles in the paper, we renamed this section simply as ‘’Results’’.
Point 6: Discussion and conclusions
The discussion seems to me to be adequate with the literature review conducted and the comparisons proposed. The conclusion should be rewritten to reinforce it and make the contributions of the paper more visible. Answer research questions posed in the introduction.
Response 6: Accepted and corrected. We changed and expanded Conclusion section.
I thank the authors for the opportunity to review their work.
Point 7: Comments on the Quality of English Language
Reduction of some long sentences to become more concrete.
Response 7: Accepted and corrected, wherever we noticed. The paper will undergo extra proofreading after acceptance.
Reviewer 2 Report
The topic of this paper is interesting and has some research value. However, the author does not seem to have mastered the methodology of an academic paper, for example, the research question is not well-presented in the introduction and the significance of the research question is not explained; the research review is not fully focused on the research question and is placed in a theoretical framework that needs to be adjusted and further focused on the research question; the theoretical framework needs to be further constructed; the area of study is described too much and the reasons for choosing this area need to be explained rather than the theoretical framework needs to be further constructed. These problems make the article difficult for the reader to understand. A complete revision is needed to make it more scientifically sound.
Author Response
Dear Reviewer 2,
First of all, thank You for your effort to review our paper and useful comments and remarks. You gave us very clear and detailed inputs how to improve this article and we really appreciate your help. Sincerely, we hope that we understood everything correctly and accomplished expectations. We tried to combine all remarks form 3 reviews, and all corrections and additions are implemented in new version of the paper, that we are uploading as an attachment.
The topic of this paper is interesting and has some research value.
However, the author does not seem to have mastered the methodology of an academic paper, for example, the research question is not well-presented in the introduction and the significance of the research question is not explained; the research review is not fully focused on the research question and is placed in a theoretical framework that needs to be adjusted and further focused on the research question; the theoretical framework needs to be further constructed; the area of study is described too much and the reasons for choosing this area need to be explained rather than the theoretical framework needs to be further constructed. These problems make the article difficult for the reader to understand. A complete revision is needed to make it more scientifically sound.
Response: Accepted and corrected. Generally, we changed a structure of the paper, moved some parts to other sections and made additions regarding research questions, theoretical framework, introductory part and conclusions. We followed remarks form Reviewers 1 and 3, because they gave as very precise guidance, about same sections and topics that You mentioned in Your comments.
Reviewer 3 Report
Dear Authors,
The article, titled "Instruments for implementing sustainable tourism development on reclaimed land in the Middle Danube River," aims to study the relationship and interaction between urban and spatial planning and tourism development, on two spatially similar case studies, analyzing the conditions, similarities and differences between them. and similarities and differences between them.
The research topic undertaken sustainable development in tourism is very important for spatial planning and maintaining balance in nature. Smart and proper spatial management benefits both the tourism industry and nature.
After reading the paper, I have the following comments and suggestions for improvement:
Structure of the article.
I suggest improving the structure of the article according to the guidelines of the journal.
A new numbering of chapters should be introduced.
1. Introduction
2. literature review/Theoretical background
3. Materials and methods
4. Results
5. Disscusion
5. Conclusion
Abstract
I propose to improve to make it more readable. I propose to improve the abstract according to the Journal "Sustainability". There is no information about the methods used and the results of the study are not presented.
1 Introduction
This chapter needs significant improvement. Important introductory information related to sustainability in tourism is missing. I propose to describe the sustainability policy in the EU and its relation to the research area. In my opinion, it should be expanded to include the following news: why was this research undertaken? What research has been done so far, where? What conclusions have been drawn from these studies. Is this article a continuation of those conclusions, or is it based on your own observations?
Include the purpose of the research and the research questions in the conclusion of the chapter.
2 Methods and Materials.
A diagram of the research procedure is missing. Please describe in more detail the case study research procedure used. "For each case study, the characteristics of the location, in terms of nature conservation, were reviewed, and then various conditions, constraints, benefits and opportunities were synthesized for the design of representative and specific tourism facilities."
3 Theoretical Framework.
Since the article deals with both urban planning and architecture, as well as tourism and nature, the literature should also address all of these elements.
4 Area of study.
I propose to move this section to Chapter 2 Methods and Materials
5 Analysis of places and possibilities of forming tourism products within them
This chapter is well presented and described. Photographs and forms of coastal development are interesting.
This section should still answer the question: what tangible benefits has this study brought to the development of sustainable tourism. This would contribute to a high improvement of this paper. The authors should compare their project and results with results from similar conducted research on this topic from other parts of Europa and all around the world.
Technical errors.
[39] - unnecessary pause
[108] - errors in notation [1] (14)
All in all, I recommend this paper for publication in the Journal “Buildings ” after major changes.
Kind regards
Author Response
Dear Reviewer 3,
First of all, thank You for your kind words, effort to review our paper and extremely useful comments and remarks. You gave us very clear and detailed inputs how to improve this article and we really appreciate your help. Sincerely, we hope that we understood everything correctly and accomplished expectations. We tried to combine all remarks form 3 reviews, and all corrections and additions are implemented in new version of the paper, that we are uploading as an attachment.
Dear Authors,
The article, titled "Instruments for implementing sustainable tourism development on reclaimed land in the Middle Danube River," aims to study the relationship and interaction between urban and spatial planning and tourism development, on two spatially similar case studies, analyzing the conditions, similarities and differences between them. and similarities and differences between them.
The research topic undertaken sustainable development in tourism is very important for spatial planning and maintaining balance in nature. Smart and proper spatial management benefits both the tourism industry and nature.
After reading the paper, I have the following comments and suggestions for improvement:
Point 1: Structure of the article.
I suggest improving the structure of the article according to the guidelines of the journal. A new numbering of chapters should be introduced.
- Introduction
- literature review/Theoretical background
- Materials and methods
- Results
- Discussion
- Conclusion
Response 1: Accepted and corrected, paper has new order with some new subtitles.
Point 2: Abstract
I propose to improve to make it more readable. I propose to improve the abstract according to the Journal "Sustainability". There is no information about the methods used and the results of the study are not presented.
Response 2: Accepted and corrected. We changed Abstract section.
Point 3: 1 Introduction
This chapter needs significant improvement. Important introductory information related to sustainability in tourism is missing. I propose to describe the sustainability policy in the EU and its relation to the research area. In my opinion, it should be expanded to include the following news: why was this research undertaken? What research has been done so far, where? What conclusions have been drawn from these studies. Is this article a continuation of those conclusions, or is it based on your own observations?
Include the purpose of the research and the research questions in the conclusion of the chapter.
Response 3: Accepted and corrected. We changed and expanded Introduction section. In brief we made connection with EU strategy for sustainable tourism principles. We expanded Introduction and Conclusion sections. The article is not a continuation of previous researches (we treat them as theoretical base and starting point for defining terms, processes, frameworks…), it is our own observation based on two examples that we found comparable, with a purpose for dissemination of findings.
Point 4: 2 Methods and Materials.
A diagram of the research procedure is missing. Please describe in more detail the case study research procedure used. "For each case study, the characteristics of the location, in terms of nature conservation, were reviewed, and then various conditions, constraints, benefits and opportunities were synthesized for the design of representative and specific tourism facilities."
Response 4: Accepted and corrected. We added diagram of the research procedure (Figure 1. Illustration of the methodical workflow), as well as more detailed description and explanation in subsection 3.1.
Point 5: 3 Theoretical Framework.
Since the article deals with both urban planning and architecture, as well as tourism and nature, the literature should also address all of these elements.
Response 5: Accepted and corrected. We searched, reviewed and selected as addition several relevant references [38-50] about nature, biodiversity, protection, planning process and architecture, in context of Danube Region and sustainable tourism.
Point 6: 4 Area of study.
I propose to move this section to Chapter 2 Methods and Materials
Response 6: Accepted and corrected.
Point 7: 5 Analysis of places and possibilities of forming tourism products within them
This chapter is well presented and described. Photographs and forms of coastal development are interesting. This section should still answer the question: what tangible benefits has this study brought to the development of sustainable tourism. This would contribute to a high improvement of this paper. The authors should compare their project and results with results from similar conducted research on this topic from other parts of Europa and all around the world.
Response 7: Accepted and corrected, partially. Unfortunately, we did not manage to make comparison with other researches, because we didn’t find any similar by location, topic or purpose (not saying they don’t exist, but in this time given for correction and scope of paper it was not possible). Sincerely hopping this paper will be useful in future for such comparison with other studies which will follow.
Point 8: Point Technical errors.
[39] - unnecessary pause
[108] - errors in notation [1] (14)
Response 8: Accepted and corrected. [1] (14) represents [number of references] and (page where extract of citation belongs), marked according to the instructions ‘’For embedded citations in the text with pagination, use both parentheses and brackets to indicate the reference number and page numbers; for example [5] (p. 10), or [6] (pp. 101–105).’’ We added letter ‘’p’/pp’.
Round 2
Reviewer 1 Report
I would like to congratulate and thanking the authors for their efforts in implementing the advice I gave in the first revision in an appropriate way. I find the current version more attractive for the academic field.
As for the introduction, the justification can be perceived, and the study has gained much more relevance. The literature review is also much improved. In terms of methodology, the effort to contrast the methods in the existing literature is appreciated and adequately supports the study. The conclusion has also gained strength and I find it much more interesting than before.
I thank the authors for the opportunity to review their work.
Author Response
Thank You for kind and supporting words, it was pleasure to follow Your remarks and we really appreciate Your wiliness to help as to improve our paper.
Reviewer 2 Report
Congratulations to the author, the article has been completely revised. The author has provided a lot of information to make the article more informative. The conclusions are also clearer. However, the structure of the essay still needs to be adjusted.
Firstly, the background and significance of the study need to be further presented at the beginning of the introduction.
Second, the reasons for selecting the cases should be presented in the research design rather than in the introduction.
Thirdly, the objectives of the study appear several times in the introduction and need to be modified.
Authors are invited to read further relevant articles and learn how to write the introduction for their publication.
Author Response
Thank You for the second round of review. We made some corrections in text according to Your comments, removing, moving or adding some sentences. Since Introduction was tailored by very precise comments of two other reviews and they are satisfied with what has achieved, so we did not have the opportunity to introduce more changes. Thank you for kind advise how to learn about writing technics.
Reviewer 3 Report
The article has been revised according to the reviewer's instructions.
In the Introduction, only one literature item is marked, while in line 148
'ber of scientific papers about river cruising industry [9-32].
as many as 23 items. Is this an incorrect entry?
It is hard to believe that 23 items were assigned to one sentence.
Similarly, below "itage along its course [2, 14-20]".
Misspellings in line 51.
Author Response
Thank you again for Your time and afford to do the second round of review.
In the Introduction, only one literature item is marked, while in line 148 'ber of scientific papers about river cruising industry [9-32]. as many as 23 items. Is this an incorrect entry? It is hard to believe that 23 items were assigned to one sentence.
Sorry, this is technical mistake in typing, corrected to [9-13]. In this group of references, we concentrated papers exclusively about river cruising industry.
Similarly, below "itage along its course [2, 14-20]".
This is correct, we listed authors and specificities of papers within the main topic of Danube basin, with the aim of pointing out the quantity, continuity and excellence of research in this area, with citation that follows.
Misspellings in line 51. Corrected.